# An assessment of factors for the cruise port of call selection: The modified fuzzy Analytic Hierarchy Process

**Thang Quyet Nguyen[1], Quynh Manh Doan [2,3], Lan Thi Tuyet Ngo [2,3]** *

**1** Faculty of Tourism & Hospitality Management, HUTECH University, Ho Chi Minh City, Vietnam, **2** Division of Economics - Social Science, Dong Nai Technology University, Bien Hoa City, Vietnam, **3** Faculty of Economics - Management, Dong Nai Technology University, Bien Hoa City, Vietnam

* ngothituyetlan@dntu.edu.vn

**Data Availability Statement:** All relevant data are within the paper and its Supporting information files.

## Abstract

Improving cruise ports of call is essential for enhancing the overall cruise experience for passengers, promoting tourism, and supporting the economic development of the regions served by these ports. Therefore, this article aims to assess selection factors (SFs) for the cruise port of call from the perspective of cruise operators (COs) and port operators (POs). In doing so, this paper first identifies SFs for the cruise port of call and establishes their hierarchical structure thanks to the extensive literature and expert brainstorming. Afterwards, The Modified Fuzzy Analytic Hierarchy Process (MFAHP) is developed to gauge the discrepancy in SFs between COs and POs. Empirical results from MFAHP pinpoint two significant SFs for POs to improve and attract their customers: customs, immigration control and quarantine (CIQ), and incentive measures. Besides, theoretical and managerial implications, and potential limitations for the next research are discussed.

## Introduction

Cruise tourism has grown dramatically until the COVID-19 pandemic. According to the official statistics from CLIA [1], the number of individual travelers cruising globally in 2019 reaches approximately 30 million people, almost doubling in quantity compared to a decade ago. Also, cruising activities sustained 1.2 million jobs, which equaled $50.24 billion in income and $150 billion in total output worldwide in 2018. Further, with an average growth rate of 6.63% in annual passengers during 1990–2020 [2], cruise liners always upgrade their navigational capacities to satisfy customers' strong demand through some solutions, such as increasing business days [3], expanding shipping routes [4], and enlarging vessel sizes [5, 6]. Cruise liners are also offering bite-sized cruises over three-to-five-day itineraries to many destinations [1]. It is argued that the recent growth of the cruise tourism industry may bring a golden opportunity for cruise ports to attract cruise liners and their passengers to generate revenues and profitability.

The expansion of cruise tourism has intensified competition among cruise ports, prompting cruise liners to choose ports as homeports, ports of call, or hybrid ports. A homeport, as defined by Marti [7], serves as the ship's registered or permanent base, where passengers

**Funding:** The author(s) received no specific funding for this work.

**Competing interests:** The authors have declared that no competing interests exist.

**Abbreviations:** CI, Consistency Index; CLIA, Cruise Lines International Association; COs, Cruise Operators; CR, Consistency Ratio; $CW_i$, COs expectation weight for the $i^{th}$ SF; MCDM, Multi-criteria decision making; PE, Port environment; PF, Port travel features; PM, Port Management; POs, Port Operators; PT, Inland transportation and port traffic; $PW_i$, POs expectation weight for the $i^{th}$ SF; rARPACK, The package in RStudio is used to compute $\lambda_{max}$; RI, Randomized index; SFs, Selection factors; SG-NP, Saigon Newport Corporation; TFN, Triangular fuzzy number; $\lambda_{max}$, The maximum eigenvalue of an individual positive reciprocal matrix.

embark and disembark. Ports of call are intermediate stops during cruise itineraries, typically lasting a few hours before continuing to another destination. Hybrid ports combine elements of both. Attracting cruise ships to call at a port generates revenue and fosters economic development for the port city. Cruise passengers spend $385 in port cities before boarding and $100 at each port destination during a voyage [1]. Despite this economic potential, factors influencing cruise port selection remain undocumented, particularly in developing countries like Vietnam.

Various policies have been proposed for cruise ports to enhance operational performance and competitiveness. Hsu, Lian [8] stress the importance of port infrastructure and service diversification, while Nguyen, Ngo [9] focus on tourism attractiveness and proximity to attraction sites. Recommendations include diversifying onshore tourism programs and securing government policy support [5]. Gouveia and Eusébio [10] advocate for local businesses' involvement through innovative approaches. However, with limited empirical comparisons, existing studies often rely on perspectives from cruise operators (COs) or port operators (POs). Notably, the proposed recommendations lack alignment with the interests of all relevant parties, including POs and port stakeholders.

Moreover, the selection of cruise ports is contended to heavily depend on the subjective judgments and multiple criteria assessments made by decision-makers, commonly known as multiple-criteria decision analysis (MCDA). In practical terms, issues related to MCDA are often addressed using the fuzzy Analytic Hierarchy Process (AHP), DEMATEL, the analytic network process (ANP), interpretive structural modelling, fuzzy decision maps, WINGS, entropy, CRITIC, SWARA, etc. Among them, fuzzy AHP is the most popular due to its straightforward algorithms. However, fuzzy AHP assumes that the criteria and sub-criteria utilized in decision-making are independent and do not interact or influence each other [8, 9, 11]. In actuality, many real-world problems, such as cruise port selection, involve dependent and interconnected criteria. Therefore, the incapacity of fuzzy AHP to consider these interdependencies may result in inaccurate conclusions and suboptimal decisions. To overcome this limitation, this study introduces the direct-effect matrix to evaluate the impact of criteria interactions on the decision outcome and to adjust the findings of fuzzy AHP appropriately.

To bridge the literature gap, this paper aims to assess COs' selection for the cruise port of call from the perspectives of COs' and POs. More especially, this article specifies the difference in SFs for the cruise port of call between COs and POs. To do so, SFs are initially identified by the literature review and the cruise ports' operational characteristics. Next, after weighting SFs by adopting the modified fuzzy AHP approach, the paper addresses the difference in SFs between COs and POs. From there, improvement policies are suggested to improve the cruise port's service quality and attract more COs. As an empirical study, the Saigon Newport Corporation (SG-NP) and its customers (i.e., COs) were surveyed to verify the proposed research model.

The rest of the paper is organized as follows: Section 2 and 3 present the literature review regarding SFs and the research method used in this study, respectively. Section 4 exhibits the empirical research results and discussion for the case study of SG-NP. Finally, we provide some conclusions, limitations, and suggestions for further research in Section 5.

## Literature review

This section attempts to disclose essential criteria for the cruise port of call (hereafter SFs) from the perspective of POs, COs, and other cruise stakeholders (i.e., travel agencies and cruise brokers).

When investigating the economics of cruising in the short-sea cruise market, Bull [12] argued that service pricing and diversification are the main factors impacting COs' destination

selection. Lee and Lee [13] explained that service price has a considerable effect on cruise passengers' budgets, while service diversification enables them to fully satisfy with the cruise tour package. Manning [14] introduced instructions for COs to successfully elect the port of call as their destination. Among the 27 explored specific SFs, the primary impacting factors are cruise port infrastructure (i.e., docking areas, bunkering, shore-side power, tugboats), the cruise port's natural and cultural heritage, the distance between destinations and the homeport, safety and security, essential provision (i.e., food and drink), the marketing strategy (i.e., the diversification of itineraries for the passenger to choose), port facilities (i.e., accessibility and convenience for COs embarking and disembarking), and port charges. McCalla [15] partitioned SFs into two types, including the site and the situation. The former refers to harbor approaches, water depth, shelter, and shore facilities influencing the viability of a port to meet the needs of cruise liners. Meanwhile, the latter regards to some factors, such as hinterland and foreland connections, and local attractions, which reflect the degree of attractiveness of the cruise port. More specifically, the situation factor might be broken down into two subfactors: the market of potential cruise passengers, and the destinations that cruise passengers and cruise ships want to come to. Other studies have also posited that the latter is more necessary [16–19].

In line with the findings of McCalla [15], Lekakou, Pallis [16] broke down SFs into site and situation requirements in the research context of the European cruise industry. In particular, the former includes natural port characteristics, port management, port infrastructure, port services for passengers and COs, port charges, city tangible amenities, the political system, port efficiency, and the regulatory framework. In the meantime, the latter consists of intermodal transportation, attraction sites around ports, and the proximity of markets for cruise passengers. Moreover, the empirical findings demonstrate that "situation" and "site" have the same significance. By applying the integration of extended VIKOR and Analytic Network Process under a decisive fuzzy environment, Demirel and Yucenur [17] unveiled four main dimensions affecting cruise selection for the port of call: strategic conditions (proximity of tourist attraction and cruise tourism markets, and expansion capabilities), technical issues (climate and weather, wharf depth, dock convenience, and wave effects), economic elements (initial investment costs, operational costs, profitability, and marketing costs), and social factors (transportation network and the local educational system).

Further, local and regional land-based attractions at cruise ports of call are crucial factors in appealing to cruise companies. It is advised that for the luxurious cruise market, cruise liners pay more attention to natural attractions, cultural shore excursions, and traditional activities at locals for cruise passengers to experience and enjoy [20, 21]. It is posited that cruise itineraries substantially influence cruise port selection, in part because they significantly impact the occupancy rate of the cruise vessel [22]. Besides, the itinerary system is seen as the basis of cruise tourism since it attracts and keeps cruise passengers' interest [23]. Casado-Díaz, Navarro-Ruiz [24] also have the same remark. On top of that, some main challenges faced by cruise liners in developing cruise itineraries include diversification and attractions at cruise ports of call [25], the distance between the homeport and the port of call [25], destination infrastructure [16], port facilities [10], security and safety [26], environmental policies [20], provisions and cleanliness [25], cruise length [13], fuel consumption and port taxes [27], and climate condition [28]. It is also ascertained that port-service providers should strengthen cruise operations' flexibility [29], stabilize political status [17], and diversify a range of tourism products [13] to service enjoyable coastal ecotourism events for cruise passengers, thereby fascinating COs.

For developing a dedicated cruise terminal, Lee [30] emphasized the significance of four kinds of facilities, including port-related facilities (docking facilities, energy supply facilities, and search and rescue systems), tourist-providing facilities (hotels, shopping centers, and

recreational areas), information technology facilities (communication networks, and telecommunications infrastructure), and customs, immigration, and quarantine (CIQ) facilities. Additionally, to enhance a cruise port's attractiveness, cruise vessels should be serviced with: basic supplies [24], maintenance and repair services [10], shops and foreign exchange bureaus [31], and tourism information offices [32]. The port government's organizational policies applying to the cruise industry are also imperative to grow the cruise tourism industry [33]. Furthermore, relaxed visa requirements [34], expedited clearance procedures [35], passenger shipping regulations [27], and taxation systems [22] are illustrated as decisive factors for COs to select a cruise port of call.

To sum up, although the preceding literature has contributed pragmatically and theoretically towards cruise port selection, some limitations should be considered. Firstly, when mentioning cruise port selection, a large fraction of prior research only leans on either POs' judgment or COs' opinion. As a result, little literature evaluates cruise port selection from multi-stakeholder perspectives. Secondly, it is believed that POs and COs are two key bodies involving the procedure for cruise port selection; thus, the difference in their cognition regarding SFs is anticipated to be a valuable source of information for POs to enhance the cruise port's performance and attract more COs to use port services. Accordingly, this article attempts to bridge the literature gap by assessing SFs from POs and COs' perspectives. Most importantly, the application of the MFAHP approach helps to determine the discrepancy in expected service demands among COs and POs. So, proposed recommendations might reflect their service requirements.

## Methods

### Hierarchical structure

To verify SFs and establish their hierarchical structure used in the modified fuzzy AHP, we utilize brainstorming, which is defined as a problem-solving method for generating creative ideas in a group setting. To do so, we initially created a summarized list of 29 critical factors impacting COs' port selection via the extensive literature review and the cruise industry's operational features. Keep in mind that such 29 factors are excluded from double counting some factors (because one factor can be referred to by a different name or expression). A pool of five worldly experts were then invited to decide which critical factors (i.e., SFs) were most crucial for COs' port selection and even found out new factors if missing. As a result, 16 most-important factors were identified, and their hierarchical structure was also established, as represented in Table 1.

### Sampling

Utilizing the MFAHP approach, this study developed a nine-point expert questionnaire to assess the relative significance of SFs from the perspectives of COs and POs. With guidance from Saigon Newport Corporation, an initial pool of 52 potential experts (30 from COs and 22 from POs) was identified for possible interviews. After thoroughly examining their backgrounds, 32 experts meeting specific criteria related to work experience and position were selected for interviews. Subsequently, invitations were delivered to these 32 experts, of which 24 agreed to participate. The research team conducted face-to-face interviews with these experts at their residences or offices. To ensure the reliability of the survey results, responses were subjected to consistency testing using Eqs (1) and (2).

In theory, the application of the fuzzy AHP approach starts by establishing individual positive reciprocal matrixes (IPRMs) through experts' ratings. Saaty and Tavana [36] suggested using consistency index (CI) and consistency ratio (CR) to test the consistency of IPRMs, as

**Table 1. SFs for the cruise port of call selection.**

| Layer 1: Dimensions | Code | Layer 2: SFs |
|---|---|---|
| Port environment (PE) | PE1 | The geographical location of ports |
| | PE2 | Sanitary conditions of local residents |
| | PE3 | Local city image (fame) |
| | PE4 | Political Factors |
| Inland transportation and Port traffic (PT) | PT1 | Connectivity between ship berthing facilities and hinterland. |
| | PT2 | Convenient transportation for inland tourism mass rapid transit system |
| | PT3 | The port is close to the international airport |
| | PT4 | Intensity of aircraft flights and route planning |
| Port travel features (PF) | PF1 | The diversity of land tourism projects |
| | PF2 | Local historical places |
| | PF3 | Modern tourist centre |
| | PF4 | Large-scale entertainment events held locally |
| Port Management (PM) | PM1 | Visa-free provision for cruise passengers |
| | PM2 | Customs, immigration control and quarantine (CIQ) |
| | PM3 | Incentive measures for cruise ship calling operations |
| | PM4 | Sea rescue system specification |

follows:

$$CR(n) = \frac{CI(n)}{MRCI(n)} \tag{1}$$

And

$$CI(n) = \frac{\lambda_{\max} - n}{n - 1} \tag{2}$$

Where $\lambda_{max}$ is the maximum eigenvalue of IPRMs, $n$ is the number of criteria in the matrix, and RI represents a randomized index, whose values are derived by undertaking the simulation experiment with 2500 runs, as shown in Table 2. In practice, the $CR < 10\%$ is an acceptable range [36].

In this paper, the software package 'rARPACK' in the RStudio was first used to find $\lambda_{max}$. And CI and CR were then obtained by Eqs (1) and (2). As a result, four questionnaires did not satisfy $CR < 10\%$; thus, we made a phone call to corresponding respondents to re-interview till their responses reached consistency. Finally, we had 24 official responses for the next analysis. By the way, respondents' backgrounds are shown in Table 3. Evidently, their seniority and job title can guarantee that they are highly competent at the assessment of criteria for the cruise port of call.

### Modified fuzzy AHP approach

As mentioned earlier, the present article develops the modified fuzzy AHP approach to consider interaction among criteria (i.e., SFs). The basic idea of this approach is that the weight of

**Table 2. Random indexes.**

| $n$ | 3 | 4 | 5 | 6 | 7 | 8 | 9 | 10 | 11 | 12 |
|---|---|---|---|---|---|---|---|---|---|---|
| RI | 0.525 | 0.882 | 1.115 | 1.252 | 1.341 | 1.404 | 1.452 | 1.484 | 1.513 | 1.535 |

**Table 3. Respondents' background.**

| Features | Range | COs | | POs | |
|---|---|---|---|---|---|
| | | Frequency | % | Frequency | % |
| Age | 41–50 | 8 | 57.14 | 8 | 80 |
| | 51–60 | 2 | 14.29 | 2 | 20 |
| | Over 61 | 4 | 28.57 | 0 | 0 |
| Education | College | 10 | 71.43 | 8 | 80 |
| | Master or above | 4 | 28.57 | 2 | 20 |
| Seniority | 10~15 | 10 | 71.43 | 8 | 80 |
| | 16–20 | 2 | 14.29 | 2 | 20 |
| | Over 21 | 2 | 14.29 | 0 | 0 |
| Job title | Manager | 8 | 57.14 | 6 | 60 |
| | Senior staff | 6 | 42.86 | 4 | 40 |

a specific criterion includes two components: the original weight and the affected weight. In particular, the affected weight is defined as the extent to which a criterion affects others. From such an idea, the conventional fuzzy AHP is first used to determine the original weight of SFs. Then, a direct-effect matrix is developed to revise the original weight.

This paper uses four criteria of the PE dimension, including PE1, PE2, PE3, and PE4, to illustrate the adoption of the modified fuzzy AHP approach in calculating SFs' weights from COs and POs' viewpoints. The application of the modified fuzzy AHP is undertaken as follows:

**Step 1**: The combination of experts' evaluations

This current paper applies fuzzy theory to measure experts' subjective evaluation. Let a triangular fuzzy number (TFN) $\tilde{A} = (l, m, u)$, then its membership function is defined by:

$$\mu_{\tilde{A}}(x) = \begin{cases} (x - l)/(m - u) & , if\ x \in [l, m] \\ (x - u)/(m - u) & , if\ x \in [m, u] \\ 0 & , if\ otherwise \end{cases} \quad (3)$$

Where parameters $l$, $m$, and $u$ stand for the lower limit, the mode, and the upper limit in the framework of fuzzy logic and fuzzy systems, respectively. When $l = m = u$, $\tilde{A}$ will degenerate into a crisp number $A$ [11].

**Step 2**: The integrated fuzzy positive reciprocal matrix (IFPRM)

Suppose that we have $n$ criteria (i.e., SFs) needing to be judged, and $\tilde{a}_{ij} = \left[ l_{ij}, m_{ij}, u_{ij} \right]; (i, j = 1, 2, \ldots, n)$ be the TFN, then IFPRM can be described as follows:

$$\tilde{A} = \left[ \tilde{a}_{ij} \right] = \begin{bmatrix} 1 & \tilde{a}_{12} & \cdots & \tilde{a}_{1n} \\ 1/\tilde{a}_{12} & 1 & \cdots & \tilde{a}_{2n} \\ \vdots & \vdots & \ddots & \vdots \\ 1/\tilde{a}_{1n} & 1/\tilde{a}_{2n} & \cdots & 1 \end{bmatrix} \quad (4)$$

This study adopts the geometric mean to combine the multi-experts' IPRMs into an IFPRM. Let $a_{ij}^k$, $i, j = 1, 2, \ldots, n$ and $k = 1, 2, \ldots, h$ be the degree of relative importance assigned

to any two criteria $i$ and $j$ by the $k^{th}$ expert. Then IPRMs formed from the $h$ experts can be integrated as:

$$\tilde{A} = \left[\tilde{a}_{ij}\right] = \left[\min_{1\leq k\leq h}\left\{a_{ij}^1, a_{ij}^2, \ldots, a_{ij}^k\right\}, \sqrt[n]{\left(\prod_{k=1}^{h} a_{ij}^k\right)}, \max_{1\leq k\leq h}\left\{a_{ij}^1, a_{ij}^2, \ldots, a_{ij}^k\right\}\right] \quad (5)$$

Applying Eq (5), the IFPRM of the PE dimension can be formed as follows:

$$\tilde{A}_1 = \begin{bmatrix} (1.000\ 1.000\ 1.000) & (1.000\ 1.246\ 2.000) & (2.000\ 2.331\ 3.000) & (2.000\ 2.746\ 3.000) \\ (0.500\ 0.803\ 1.000) & (1.000\ 1.000\ 1.000) & (1.000\ 2.322\ 4.000) & (1.000\ 2.169\ 6.000) \\ (0.333\ 0.429\ 0.500) & (0.250\ 0.431\ 1.000) & (1.000\ 1.000\ 1.000) & (1.000\ 1.443\ 3.000) \\ (0.333\ 0.364\ 0.500) & (0.167\ 0.461\ 1.000) & (0.333\ 0.693\ 1.000) & (1.000\ 1.000\ 1.000) \end{bmatrix}$$

**Step 3**: The IFPRM's consistency

According to Wang and Lin [37], IFPRMs can be checked for consistency by the geometric consistency index (*GCI*). Let $\tilde{A} = \left(\tilde{a}_{ij}\right) = \left(a_{ij}^L, a_{ij}^M, a_{ij}^U\right)_{n\times n}$ be IFPRMs, whose *GCI* is Figd out as:

$$GCI(\tilde{A}) = \max\left\{\frac{2}{(n-1)(n-2)}\sum_{i<j}\left(\ln a_{ij}^M - \frac{1}{n}\sum_{k=1}^{n}\ln a_{ik}^M + \ln a_{kj}^M\right)^2;\right.$$
$$\left.\frac{1}{2(n-1)(n-2)}\sum_{i<j}\left[\ln a_{ij}^L + \log a_{ij}^U - \frac{1}{n}\sum_{k=1}^{n}\left(\ln a_{ik}^L + \ln a_{ik}^U + \ln a_{kj}^L + \ln a_{kj}^U\right)\right]^2\right\} \quad (6)$$

It is argued that the *GCI* thresholds rely on the number of criteria IFPRMs' criteria. In particularly:

$$GCI = \begin{cases} 0.3147 & \text{if } n = 3 \\ 0.3562 & \text{if } n = 4 \\ 0.3700 & \text{if } n > 5 \end{cases}$$

Revert to the matrix $\tilde{A}_1$, its GCI is computed as: $GCI(\tilde{A}_1) = \max\{0.110; 0.077\} = 0.110$. It is evident that the matrix $\tilde{A}_1$ is consistent. Similarly, the consistency test for remaining IFPRMs of the SG-NP case is demonstrated in Table 4.

**Step 4**: Original weight of SFs

Until now, there have been various methods to compute the priority weight of IFPRMs, such as fuzzy extent analysis [38], the centroid method [39], the fuzzy geometric means of the rows [40], logarithmic least squares [41], etc. This article adopts the fuzzy geometric means of

**Table 4. Consistency of IFPRMs.**

| | Layer | GCI | Threshold | Result |
|---|---|---|---|---|
| COs expectation | Layer 1 | 0.225 | 0.3562 | Consistent |
| | Layer 2: PE | 0.252 | 0.3562 | Consistent |
| | Layer 2: PT | 0.179 | 0.3562 | Consistent |
| | Layer 2: PF | 0.342 | 0.3562 | Consistent |
| | Layer 2: PM | 0.086 | 0.3562 | Consistent |
| POs expectation | Layer 1 | 0.121 | 0.3562 | Consistent |
| | Layer 2: PE | 0.110 | 0.3562 | Consistent |
| | Layer 2: PT | 0.243 | 0.3562 | Consistent |
| | Layer 2: PF | 0.208 | 0.3562 | Consistent |
| | Layer 2: PM | 0.334 | 0.3562 | Consistent |

the rows to determine the priority weight of IFPRMs as follows:

$$\tilde{w}_i = \left[\frac{\left(\prod_{j=1}^{n} l_{ij}\right)^{1/n}}{\sum_{i=1}^{n}\left(\prod_{j=1}^{n} u_{ij}\right)^{1/n}}, \frac{\left(\prod_{j=1}^{n} m_{ij}\right)^{1/n}}{\sum_{i=1}^{n}\left(\prod_{j=1}^{n} m_{ij}\right)^{1/n}}, \frac{\left(\prod_{j=1}^{n} u_{ij}\right)^{1/n}}{\sum_{i=1}^{n}\left(\prod_{j=1}^{n} l_{ij}\right)^{1/n}}\right]; i = 1, 2, \ldots, n \quad (7)$$

Finally, based on Eq (7), we can find the weight matrix for the $i^{th}$ SF ($i$ = 1,2,. . .,4) as:

$$\tilde{w} = \begin{bmatrix} 0.227 & 0.382 & 0.652 \\ 0.135 & 0.322 & 0.700 \\ 0.059 & 0.133 & 0.266 \end{bmatrix}$$

Next, we can calculate the graded mean integration representation (GMIR) of the matrix $\tilde{w}$:

$$w_i = \frac{l_i^w + 4 \times \left(m_i^w\right) + u_i^w}{6}, i = 1, 2, \ldots, n \quad (8)$$

Normalize $w_i(i = 1, 2, \ldots, n)$, we obtain the crisp weight of the $i^{th}$ SF:

$$\omega_i = \frac{w_i}{\sum_{i=1}^{n} w_i}, i = 1, 2, \ldots, n \quad (9)$$

Be means of Eqs (8) and (9), the original weight of the PE dimension (i.e., PE1, PE2, PE3, and PE4) can be obtained as follows:

$$w^{GE} = \begin{bmatrix} 0.4010 \\ 0.3540 \\ 0.1816 \\ 0.1427 \end{bmatrix} => \omega^{GE} = \begin{bmatrix} 0.3716 \\ 0.3280 \\ 0.1682 \\ 0.1322 \end{bmatrix}$$

By the same way, the original weight of the remaining dimensions can be determined and shown in Table 5.

**Step 5**: Revising SFs' original weight

**Table 5. SFs' original weight for COs and Pos.**

| Layer 1: Dimensions | Global weight in Layer 1 (%) | | Layer 2: SFs | Local weight in Layer 2 (%) | |
| --- | --- | --- | --- | --- | --- |
| | COs | POs | | CO2 | POs |
| PE | 19.63 | 16.69 | PE1 | 30.48 | 37.16 |
| | | | PE2 | 18.58 | 32.80 |
| | | | PE3 | 21.15 | 16.82 |
| | | | PE4 | 29.80 | 13.22 |
| PT | 15.39 | 29.01 | PT1 | 28.94 | 15.34 |
| | | | PT2 | 18.38 | 10.35 |
| | | | PT3 | 22.43 | 23.22 |
| | | | PT4 | 30.26 | 51.09 |
| PF | 16.44 | 22.58 | PF1 | 27.33 | 24.76 |
| | | | PF2 | 16.88 | 10.71 |
| | | | PF3 | 24.62 | 25.95 |
| | | | PF4 | 31.17 | 38.57 |
| PM | 48.54 | 31.72 | PM1 | 30.73 | 21.88 |
| | | | PM2 | 16.00 | 10.24 |
| | | | PM3 | 30.28 | 23.06 |
| | | | PM4 | 22.99 | 44.83 |

To consider the mutual relation among the SFs, a direct-effect matrix is developed in this paper to revise the SFs' original weights, as done in Steps 1~4. The revision process is then implemented to examine the interrelationship among criteria within one dimension. In this paper, the revision process is employed through three main sub-steps, as below:

(1) The definition of the direct-effect matrix

For $n$ SFs, its direct-effect matrix ($M$) is symbolized as:

$$M = \left[ m_{ij} \right]_{n \times n}, i, j = 1, 2, \ldots n \tag{10}$$

In which: the $m_{ij}$ is the extent that $SF_i$ affects $SF_j$, or conversely the extent that $SF_j$ is affected by $SF_i$. Furthermore, the degree a particular SF affects itself should be 100%, which is already measured by its original weight ($\omega_i$, $i = 1, 2, \ldots, n$). Thus, it is not considered in the direct-effect matrix (i.e., $m_{ij} = 0$ for $i = j$). In other words, the diagonal $m_{ij}$ in the matrix $M$ will be 0. In this paper, a Likert scale was used to measure the $m_{ij}$, which is scored from 0 (no effect degree) to 4 (high-effect degree).

As in the previous section, we still take SFs (i.e., PF1, PF2, PF3, and PF4) under the PF dimension as an example to explain how to construct the direct-effect matrix. In this article, the value $m_{ij}$ is obtained by surveying respondents, as discussed in Section 3.3. Subsequently, an arithmetic mean is used to average respondents' scores: $M = \left( \sum_{k=1}^{h} m_{ij}^h \right) / h$. As a result, the direct-effect matrix for the PF dimension is found as:

$$M_1 = \begin{bmatrix} m_{11} & m_{12} & m_{13} & m_{14} \\ m_{21} & m_{22} & m_{23} & m_{24} \\ m_{31} & m_{32} & m_{33} & m_{34} \\ m_{41} & m_{42} & m_{43} & m_{44} \end{bmatrix} = \begin{bmatrix} 0 & 0.51 & 0.27 & 0.76 \\ 2.13 & 0 & 2.25 & 2.52 \\ 2.79 & 2.57 & 0 & 2.63 \\ 1.77 & 1.28 & 2.49 & 0 \end{bmatrix}.$$

In the matrix $M_1$, it is clear that the degree $SF_1$ affects $SF_2$ is 0.51 ($m_{12} = 0.51$), while the degree $SF_2$ affects $SF_1$ is 2.13 ($m_{21} = 2.13$).

(2) The normalized direct-effect matrix

To assure the direct-effect matrix $M$ can be converged in the long-term, we normalize the matrix $M$ as:

$$T = \left[ \frac{M_{ij}}{A} \right]_{n \times n}, i = 1, 2, \ldots, n, j = 1, 2, \ldots, n \tag{11}$$

Where:

$$A = \max \left[ \max_{1 \leq i \leq n} \sum_{j=1}^{n} m_{ij}, \max_{1 \leq j \leq n} \sum_{i=1}^{n} m_{ij} \right] \tag{12}$$

Note that in Eq (12), $\max_{1 \leq i \leq n} \sum_{j=1}^{n} \max_{1 \leq i \leq n} \sum_{j=1}^{n} m_{ij}$ represents the maximum total effect $SF_i$ has on other SFs; while $\max_{1 \leq j \leq n} \sum_{i=1}^{n} m_{ij}$ stands for the maximum total effect $SF_j$ is affected by other SFs.

Revert to the PF construct, by Eq (12), we have $A = 7.99$. Thus, based on Eq (11), the direct-effect matrix $M$ can be normalized:

$$T_1 = \begin{bmatrix} 0 & 0.064 & 0.034 & 0.095 \\ 0.267 & 0 & 0.282 & 0.315 \\ 0.349 & 0.322 & 0 & 0.329 \\ 0.222 & 0.160 & 0.312 & 0 \end{bmatrix}$$

(3) The direct-effect matrix in the long term: According to the organizational learning theory, an organization could learn to adapt to changes in external environments [42]. This implies that if one $SF_i$ is influenced by another, it might gradually get weaker in the long term. From this idea, the matrix $T$ in the long term is defined by:

$$L = T + T^2 + \cdots + T^k \tag{13}$$

Once the matrix $T$ is normalized in Sub-step (2), we have $\lim_{k \to \infty} T^k = O$. Thus, Eq (13) can be rewritten as:

$$L = \lim_{k \to \infty} \left( T + T^2 + \cdots + T^k \right) = T(I - T)^{-1} \tag{14}$$

Based on Eq (14), the matrix $T$ in the long-term for the construct PF can be found:

$$L_1 = \begin{bmatrix} 0.136 & 0.152 & 0.144 & 0.203 \\ 0.740 & 0.363 & 0.629 & 0.707 \\ 0.843 & 0.640 & 0.445 & 0.758 \\ 0.633 & 0.451 & 0.583 & 0.394 \end{bmatrix}$$

**Step 6**: The final (local) weights of SFs

As noted earlier, this paper defines the final (local) weight of a SF, containing two parts: the original weight and the affected weight. Thus, according to such an idea, let $\omega = [\omega_1, \omega_2, \ldots,$

$\omega_n]^T$ and $\psi = [\psi_1, \psi_2, \ldots, \psi_n]^T$ represent the SFs' original and final weights, respectively. Then, the final weight can be found as:

$$\psi = \omega + L \times \omega \tag{15}$$

Where $L \times \omega$ is the vector of the SFs' affected weight.

For the PF construct, the original weights of SFs (i.e., PF11, PF22, PF3, and PF4), as shown in the last field of Table 5, are: (0.3716, 0.3280, 0.1682, 0.1322). By Eq (15), its final weight vector can be found as:

$$\psi = \begin{bmatrix} 0.3716 \\ 0.3280 \\ 0.1682 \\ 0.1322 \end{bmatrix} + \begin{bmatrix} 0.136 & 0.152 & 0.144 & 0.203 \\ 0.740 & 0.363 & 0.629 & 0.707 \\ 0.843 & 0.640 & 0.445 & 0.758 \\ 0.633 & 0.451 & 0.583 & 0.394 \end{bmatrix} \times \begin{bmatrix} 0.3716 \\ 0.3280 \\ 0.1682 \\ 0.1322 \end{bmatrix} = \begin{bmatrix} 0.5229 \\ 0.9213 \\ 0.8663 \\ 0.6656 \end{bmatrix}$$

Finally, we normalize $\psi$ as:

$$\varpi_i = \frac{\psi_i}{\sum\limits_{i=1}^{n} \psi_i}, 1 = 1, 2, \ldots, n \tag{16}$$

Consequently, the final (local) weight of the SFs can be normalized as:

$$\varpi = \begin{bmatrix} 0.0.1757 \\ 0.3096 \\ 0.2911 \\ 0.2237 \end{bmatrix}$$

Based on the above process, the SFs' final (local) weights in the other constructs for COs and POs can also be found in Table 6.

**Step 7**: SFs' global weight

The last work of the modified fuzzy AHP adoption is to compute SFs' global weight. Propose that $C = (C_1, C_2, \ldots, C_q, \ldots, C_Q)$ is the vector of dimensions' global weight, and $C_q = (C_{q1}, C_{q2}, \ldots, C_{qi}, \ldots, C_{qn})$ is the vector of local weights corresponding to the dimension $C_q$ Ultimately, the global weight for one $SF_i \in C_q$ might be computed by:

$$G_i = \frac{C_q \times C_{q1}}{100}; i = 1, 2, \ldots, n; q = 1, 2, \ldots, Q \tag{17}$$

By virtue of Eq (17), SFs' global weight for COs and POs are obtained and shown in the two last columns of Table 6.

## The discrepency in expectation between COs and POs

Assuming now that $CW_i$ and $PW_i$ are the COs expectation and POs expectation weights for the $i^{th}$ SFs, respectively. Then, the discrepency in expectation between COs and POs for the $i^{th}$ SF is symbolized as:

$$CGI_i = CW_i - PW_i, i = 1, 2, \ldots, n \tag{18}$$

**Table 6. SFs' revised weight for COs and Pos.**

| Layer 1: Dimensions | Global weight in Layer 1 (%) | | Layer 2: SFs | Local weight in Layer 2 (%) | | Global weight in Layer 2 (%) | |
|---|---|---|---|---|---|---|---|
| | CO2 | POs | | CO2 | POs | CO2 | POs |
| PE | 27.93 | 21.83 | PE1 | 24.69 | 17.57 | 4.91 | 4.91 |
| | | | PE2 | 33.16 | 30.96 | 8.65 | 8.65 |
| | | | PE3 | 23.94 | 29.11 | 8.13 | 8.13 |
| | | | PE4 | 18.21 | 22.37 | 6.25 | 6.25 |
| PT | 22.87 | 26.23 | PT1 | 21.36 | 29.00 | 6.63 | 6.63 |
| | | | PT2 | 29.81 | 17.88 | 4.09 | 4.09 |
| | | | PT3 | 17.07 | 25.36 | 5.80 | 5.80 |
| | | | PT4 | 31.76 | 27.76 | 6.35 | 6.35 |
| PF | 23.12 | 25.88 | PF1 | 29.15 | 24.49 | 5.66 | 5.66 |
| | | | PF2 | 19.45 | 29.29 | 6.77 | 6.77 |
| | | | PF3 | 35.78 | 32.07 | 7.41 | 7.41 |
| | | | PF4 | 15.62 | 14.15 | 3.27 | 3.27 |
| PM | 26.08 | 26.06 | PM1 | 18.07 | 27.12 | 7.07 | 7.07 |
| | | | PM2 | 33.23 | 18.89 | 4.93 | 4.93 |
| | | | PM3 | 21.01 | 31.66 | 8.26 | 8.26 |
| | | | PM4 | 27.70 | 22.34 | 5.83 | 5.83 |

Apparently, port managers should pay more attention to higher-CGI SFs to enhance port performance and attract more cruise carriers.

Based on COs expectation and POs expectation weights as exhibited in the two last columns of Table 6, the indexes representing the discrepancy in expectation between COs and POs regarding SFs are calculated and shown in the fourth column of Table 7. Moreover, the software package "Shiny" in RStudio is employed to visualize the distribution of SFs, shown in Fig 1.

**Table 7. The distance between COs and POs for the SG-NP case.**

| SFs | COs expectation weights | POs expectation weights | Distances | Distribution |
|---|---|---|---|---|
| PM2 | 8.67 | 4.93 | 3.74 | Positive-distance area |
| PT2 | 6.82 | 4.09 | 2.73 | |
| PE1 | 6.90 | 4.91 | 1.99 | |
| PM4 | 7.22 | 5.83 | 1.40 | |
| PF1 | 6.74 | 5.66 | 1.08 | |
| PT4 | 7.26 | 6.35 | 0.91 | |
| PF3 | 8.27 | 7.41 | 0.86 | |
| PE2 | 9.26 | 8.65 | 0.61 | |
| PF4 | 3.61 | 3.27 | 0.34 | |
| PE4 | 5.09 | 6.25 | -1.16 | Negative-distance area |
| PE3 | 6.69 | 8.13 | -1.44 | |
| PT1 | 4.88 | 6.63 | -1.75 | |
| PT3 | 3.90 | 5.80 | -1.90 | |
| PF2 | 4.50 | 6.77 | -2.27 | |
| PM1 | 4.71 | 7.07 | -2.36 | |
| PM3 | 5.48 | 8.26 | -2.78 | |

## Results and discussion

### The weight of selection factor for cruise operators

As seen in the second column of Table 6, COs take most notice of port environment (PE, 27.93%) when choosing a cruise port of call, followed by port management (PM, 26.08%), port travel features (PF, 23.12%), and inland transportation and port traffic (PT, 22.87%). It is confirmed that one of the primary functions of a cruise port of call is to provide cruise customers with intimate experiences, such as itinerary attractions and cultural diversification, so that they can enjoy their lives [35, 43]. Accordingly, under the customers' side, the selection of cruise tours is highly dominated by the cruise port environment. Nguyen, Ngo [9] contended that a conducive port environment, such as favorable locations and political stability, could facilitate cruise customers in fulfilling their cruise vacation desires. This, in turn, could serve as a motivating factor for COs to choose particular cruise ports. This emphasis on the port environment by COs is likely due to its significant impact on meeting the preferences and satisfaction of cruise customers.

Additionally, while port management is deemed the *necessary condition* for COs to call a cruise port, port management is regarded as the *sufficient condition*, which enforces the cruise port selection process. K Hsu, S Huang [44] demonstrated that effective port management expedites administrative procedures at cruise ports, significantly reducing the time customers spend on security screening, check-in, and check-out activities. Hence, it is crucial to consider optimizing the port management process to enhance port operational efficiency and appeal to COs.

Among 16 SFs, the sanitary conditions of local residents are gauged to be the most necessary factor for COs to call a cruise port. Remember that cruise vacations are often luxurious tours for the rich, who are believed to notice more sanitary conditions [24, 45], such as food safety and hygiene, a healthy and fresh living environment, clean drinking water, and adequate treatment. Therefore, for the case of the SG-NP, the port government and POs are suggested to propagate the overall image of the port city with cleanliness, and safety. To do so, the Vietnam National Authority of Tourism is responsible for developing the cruise tourism information system and the cross-border e-commerce platform to assist cruise tourists. Teng, Wu [46] also had a similar suggestion.

CIQ is the second—most important factor for COs to select the port of call. According to Brida, Pulina [47], cruise passengers often spend 4–8 hours visiting the port city when arriving at a cruise port of call. Thus, the processing time of CIQ is of paramount importance for COs in designing a cruise itinerary [20, 44]. A convenient CIQ procedure not only extends the staying time of cruise passengers, and cruise members in the port city [35, 48], but also boosts their level of expenditure on food, beverages, and shopping [24]. Accordingly, this paper suggests that the Vietnam National Immigration Agency simplify the entry visa application process and grant a 24-hour transit visa exemption to appeal to more cruise carriers and cruise passengers.

The third-largest factor impacting cruise port selection is the modern tourist center in the port city. In fact, cruise passengers demand to get pleasure from their cruise vacation when booking a cruise tour. Arguably, cruise passengers not only desire to explore onboard activities (i.e., spa and fitness, dining, and entertainment), but also relish onshore excursions, which are special picnics arranged by cruise companies as an extra package. Gouveia and Eusébio [10] posited that sightseeing, shopping, and eating at luxurious tourist centers in the port city could gratify cruise passengers with their cruise journey. Therefore, the government should modernize tourist centers in the port city to provide cruise customers with novel experiences, and then attracting COs to elect the port of call. Note that shopping, entertainment, and eating centers

at tourist centers must be developed to serve cruise passengers with high-quality and special-ized services.

## Analyzing the difference in COs and POs' expectation

Table 7 illustrates two groups of SFs: negative-distance SFs in Zone I and positive- distance SFs in Zone II. It is to be borne in mind that if the COs expectation weights are larger than the POs expectation weights, SFs will be positive distances. Put succinctly, port executives should turn their attention to these SFs. By contrast, if the COs expectation weight is lower than the POs expectation weight, SFs will be negative distances. This also implies that such SFs are over-invested.

Additionally, Fig 1 provides a generic picture of the scattering of SFs on a two-dimensional (2D) hyperplane. More specifically, the COs expectation weights are depicted by the horizontal axis, while the POs expectation weights are represented by the vertical axis. Furthermore, the 2D hyperplane is evenly separated into two zones by a 45-degree line. Virtually, all SFs in Zone I have COs expectation weights < POs expectation weights; so, they are evaluated as negative distances. Conversely, all SFs in Zone II have COs expectation weights > POs expectation weights; thus, they are assessed as positive distances. In addition, the scale of the distances is gauged by the gap between the SF positions in the 2D hyperplane and the 45-degree line. For instance, the PM2's coordinate is (8.67%, 4.93%), then its distance is obtained as +3.74%, dem-onstrating that it is a positive-distance SF. In the same vein, PM3 has a negative distance of −2.78%.

These findings may be convincing evidence for port managers in reallocating resources, including managerial and physical resources, to advance port operational capabilities. In prac-tical application, port managers ought to notice positive-distance SFs in Zone II. In other words, POs' governance policies to improve port operations will concentrate SFs in Zone II. Conversely, negative-distance SFs in Zone I are regarded as "relatively overkill"; so, limited resources allocated for them should be transferred elsewhere, especially to positive-distance SFs in Zone II.

However, notice that POs might not simultaneously ameliorate all positive-distance SFs in Zone II under scarce resources. Alternatively, they ought to choose a few SFs as priority

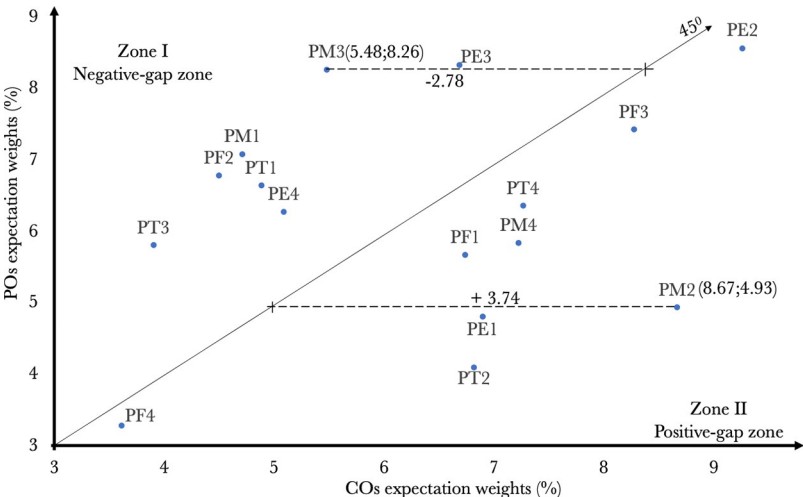

**Fig 1.**

settings, preferably SFs with a higher distance. For instance, CIQ (PM2) has the largest distance; thereby, it should be prioritized for improvement policies. The next to be considered are PT2, PE1, etc. Similarly, port executives should gradually move resources from negative-distance SFs. Evidently, incentive measures for cruise ship calling operations (PM3) have the biggest distance; hence, this criterion should be elected for resource reallocation first. The next to be taken into account are PM1, PF2, etc.

## Conclusion

For long-coastline nations, such as Vietnam, cruise tourism is growing rapidly, contributing substantially to the GDP growth rate. Undeniably, the selection of the cruise port of call from COs brings many big economic benefits, such as generating jobs, income, and the local budget. It is argued that in this dynamic cruise tourism market, what the cruise port of call should do to improve its operational quality to attract cruise vessels and passengers is crucial for the economic development of a nation. Therefore, this paper aims to assess SFs for the cruise port of call, with the Saigon Newport Corporation in Vietnam as the empirical case.

The empirical results from MFAHP show that according to COs' perspectives, the port environment is the most-interested criteria when electing a cruise port of call (27.93%), followed by port management (PM, 26.08%), port travel features (PF, 23.12%), and inland transportation and port traffic (PT, 22.87%). Furthermore, among 16 SFs, the sanitary conditions of local residents, CIQ, and the modern tourist center are assessed to be the most necessary SFs for COs to call a cruise port. From the results, some recommendations are proposed, as follows: (1) propagating the overall image of Vietnam with cleanliness, and safety; (2) developing the cruise tourism information system and the cross-border e-commerce platform to assist cruise tourists; (3) simplifying the entry visa application process and granting a 24-hour transit visa exemption; and (4) modernizing tourist centers in the port city to provide cruise customers with novel experiences.

Different from the past literature, the present article introduces the direct-effect matrix to revise the traditional fuzzy AHP, which assumes the independence of criteria. It is postulated that such an assumption is not realistic in many real-world situations. Thus, this paper contributes significantly to the development of fuzzy AHP theory in particular and MCDA in general. More importantly, this article measure the disparity in expectation between COs and POs in terms of SFs. This model also provides the necessary evidential basis for port executives to reallocate scarce resources to boost port operational capacities and attract COs. Consequently, the empirical results imply that two SFs with significant distance include: CIQ (positive-distance) and incentive measures for cruise ship calling operations (negative-distance). From these findings, several improvement policies are suggested for the SG-NP case. On top of that, determining the discrepancy in expectation between COs and POs contributes the methodological reference for research towards managerial decision-making.

Some limitations still exist in this research work. First of all, it is a challenge to include all relevant decision criteria in the research framework. Only 16 SFs in this research definitely do not reflect all of the pertinent criteria that COs are interested in; thus, it is advisable that further studies expand the number of port selection criteria by incorporating inputs from other port stakeholders, such as cruise passengers, or cruise agencies. The next is the small sample size for the SG-NP case. Particularly, this paper merely empirically surveyed 14 respondents from COs and 10 from POs due to time and resource restrictions. Nevertheless, a larger sample size would provide more reliable and robust conclusions. It is suggested that further studies should expand the number of surveyed respondents to deal with this disadvantage and improve improvement policies.

## Supporting information

**S1 File.**
(PDF)

## Acknowledgments

The authors would like to thank colleagues for very thoughtful reviews and critical comments, which have led to significant improvements to the early versions of the manuscript.

## Author Contributions

**Conceptualization:** Thang Quyet Nguyen.

**Data curation:** Thang Quyet Nguyen.

**Formal analysis:** Thang Quyet Nguyen.

**Methodology:** Quynh Manh Doan.

**Resources:** Quynh Manh Doan.

**Software:** Quynh Manh Doan.

**Validation:** Lan Thi Tuyet Ngo.

**Visualization:** Lan Thi Tuyet Ngo.

**Writing – original draft:** Lan Thi Tuyet Ngo.

**Writing – review & editing:** Lan Thi Tuyet Ngo.

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
