## [Decision Letter · Decision Letter 0]

27 Nov 2023

PONE-D-23-31432An assessment of factors for the cruise port of call selection: The cognitive-gap model based on the modified fuzzy Analytic Hierarchy ProcessPLOS ONE

Dear Dr. Ngo,

Thank you for submitting your manuscript to PLOS ONE. After careful consideration, we feel that it has merit but does not fully meet PLOS ONE’s publication criteria as it currently stands. Therefore, we invite you to submit a revised version of the manuscript that addresses the points raised during the review process.

We look forward to receiving your revised manuscript.

Kind regards,

Mercedes Castro-Nuño

Academic Editor

PLOS ONE

Journal Requirements:

4. Your abstract cannot contain citations. Please only include citations in the body text of the manuscript, and ensure that they remain in ascending numerical order on first mention.

Reviewers' comments:

Reviewer's Responses to Questions

**Comments to the Author**

1. Is the manuscript technically sound, and do the data support the conclusions?

Reviewer #1: Yes

2. Has the statistical analysis been performed appropriately and rigorously? 

Reviewer #1: Yes

3. Have the authors made all data underlying the findings in their manuscript fully available?

Reviewer #1: Yes

4. Is the manuscript presented in an intelligible fashion and written in standard English?

Reviewer #1: Yes

5. Review Comments to the Author

Reviewer #1: The research problem is interesting and helpful in assessing selection factors (SFs) for the cruise port operators. The research process is also rigorous. However, there still are some questions need to be clarified before publication.

6. PLOS authors have the option to publish the peer review history of their article (what does this mean?). If published, this will include your full peer review and any attached files.

Reviewer #1: No

---

## [Author Response · Author response to Decision Letter 0]

14 Dec 2023

The research problem is interesting and helpful in assessing selection factors (SFs) for the cruise port operators. The research process is also rigorous. However, there still are some questions need to be clarified before publication. Please consider the comments below:

(1) The definition of gap is not so appropriate that may confuse readers. I suggest the title of the manuscript delete the gap expression.

[Answer]: Thanks so much for the reviewer’s comment. We revised the manuscript title to address this suggestion.

(2) To make the paper more readable, the abstract should be presented as: introduction, purpose, method, result, and conclusion.

[Answer]: Thanks so much for the reviewer’s comment. We revised the manuscript abstract to address this suggestion.

(3) The introduction should be enriched by some following suggestions: The Introduction must present the motivations of the study from the point of view of literature gaps. At present, the build-up of the motivations, including the contributions of the study, is quite messy. It is difficult to clearly assess the gaps that are advanced in this work. For example, you mentioned the modified fuzzy AHP without reviewing other approaches to tackling the prioritization of crucial factors in cruise port selection. How does the modified fuzzy AHP work in comparison with other MCDM or problem structuring techniques, e.g., DEMATEL, ANP, interpretive structural modelling, fuzzy decision maps, WINGS, entropy, CRITIC, SWARA, among others. There is a whole lot of literature on this topic. The choice of the modified fuzzy AHP must be properly motivated. Thus, I suggest that the introduction section must be written from a general perspective. The message should encompass the need for a better understanding of crucial choice factors in selecting a cruise port based on the literature and the necessary characteristics of the problem that requires the use of an MCDM method, and eventually the AHP.

[Answer]: Thanks so much for the reviewer’s comment. We revised the introduction section to address this suggestion.

(4) In the literature review, the manuscript extensively analyses too many previous studies. However, some key articles about port selection are not incorporated. Authors should search relevant articles by keyword: “port selection” to find the key articles. Further, it will be better if references are updated to 2023.

[Answer]: Thanks so much for the reviewer’s comment. We revised the literature review section to update the newest research in terms of port selection.

(5) The research framework fails to provide a thorough and robust justification of the proposed “gap model”. What is it all about? How does it advance our understanding of the cruise port selection problem? I suggest delete Section 3.1 since it is not necessary in the manuscript.

[Answer]: Thanks so much for the reviewer’s comment. We deleted Section 3.1 as suggested.

(6) The visualization Figure 2 for the original and affected weights between CFi and CFj , is a mistake, should be SFi and SFj. In fact, the figure cannot express the whole conception of the interrelation of SFs, and may confuse the readers. The mathematical description is clear enough. Accordingly, I suggest delete it out of the manuscript.

[Answer]: Thanks so much for the reviewer’s comment. We deleted Figure 2 as suggested.

(7) It has been argued that the selection of the research sample is of paramount importance. Accordingly, the paper should elucidate how experts are selected for interview.

[Answer]: Thanks so much for the reviewer’s comment. We revised the manuscript to elucidate how experts were selected for interview.

(8) The paper mentions the term the "original weight" and the "affected weight". Please explain what they mean and how they are determined in the paper.

[Answer]: Thanks so much for the reviewer’s comment. We would like to explain, as follows:

Original weight is initial value assigned to selection factors (SFs), often in the context of a measurement or parameter. This kind of weight is determined by fuzz AHP. Meanwhile, affected weight is defined as the extent to which a criterion (i.e., SF) affects others. In our manuscript, this type of weight is estaimated via a direct-effect matrix.

(9) Discussion is well-written. But, it will be better if authors compare empirical results with what has been done in the relevant literature. By doing so, the difference and similarity can be clearer.

[Answer]: Thanks so much for the reviewer’s comment. We revised the discussion section to address this suggestion.

(10) Finally, there are some grammatical errors in the article, so it needs to be proofread by an English native speaker. Further. there are also several inconsistencies of mathematical notations throughout the manuscript.

[Answer]: Thanks so much for the reviewer’s comment. We got our manuscript froopread by an English native speaker. Besides, mathematical notations throughout the manuscript were checked.

---

## [Decision Letter · Decision Letter 1]

3 Jan 2024

An assessment of factors for the cruise port of call selection: The modified fuzzy Analytic Hierarchy Process

PONE-D-23-31432R1

Dear Dr. Lan Thi Tuyet Ngo,

We’re pleased to inform you that your manuscript has been judged scientifically suitable for publication and will be formally accepted for publication once it meets all outstanding technical requirements.

Kind regards,

Mercedes Castro-Nuño

Academic Editor

PLOS ONE

Reviewers' comments:

Reviewer's Responses to Questions

**Comments to the Author**

1. If the authors have adequately addressed your comments raised in a previous round of review and you feel that this manuscript is now acceptable for publication, you may indicate that here to bypass the “Comments to the Author” section, enter your conflict of interest statement in the “Confidential to Editor” section, and submit your "Accept" recommendation.

Reviewer #1: All comments have been addressed

2. Is the manuscript technically sound, and do the data support the conclusions?

Reviewer #1: Yes

3. Has the statistical analysis been performed appropriately and rigorously? 

Reviewer #1: Yes

4. Have the authors made all data underlying the findings in their manuscript fully available?

Reviewer #1: Yes

5. Is the manuscript presented in an intelligible fashion and written in standard English?

Reviewer #1: Yes

6. Review Comments to the Author

Reviewer #1: The revision is probably OK! Now I can suggest publishing the paper in the journal. However, a proofreading is encouraged to do again

7. PLOS authors have the option to publish the peer review history of their article (what does this mean?). If published, this will include your full peer review and any attached files.

Reviewer #1: No

---

## [Editor Report · Acceptance letter]

25 Jan 2024

PONE-D-23-31432R1 

PLOS ONE

Dear Dr. Ngo, 

I'm pleased to inform you that your manuscript has been deemed suitable for publication in PLOS ONE. Congratulations! Your manuscript is now being handed over to our production team.

Kind regards, 

on behalf of

Dr. Mercedes Castro-Nuño 

Academic Editor

PLOS ONE